# Kinetics-based inference of environment-dependent microbial interactions and their dynamic variation

Hyun-Seob Song,[1,2] Na-Rae Lee,[3] Aimee K. Kessell,[1] Hugh C. McCullough,[1] Seo-Young Park,[4] Kang Zhou,[5] Dong-Yup Lee[4]

**ABSTRACT** Microbial communities in nature are dynamically evolving as member species change their interactions subject to environmental variations. Accounting for such context-dependent dynamic variations in interspecies interactions is critical for predictive ecological modeling. In the absence of generalizable theoretical foundations, we lack a fundamental understanding of how microbial interactions are driven by environmental factors, significantly limiting our capability to predict and engineer community dynamics and function. To address this issue, we propose a novel theoretical framework that allows us to represent interspecies interactions as an explicit function of environmental variables (such as substrate concentrations) by combining growth kinetics and a generalized Lotka-Volterra model. A synergistic integration of these two complementary models leads to the prediction of alterations in interspecies interactions as the outcome of dynamic balances between positive and negative influences of microbial species in mixed relationships. The effectiveness of our method was experimentally demonstrated using a synthetic consortium of two *Escherichia coli* mutants that are metabolically dependent (due to an inability to synthesize essential amino acids) but competitively grow on a shared substrate. The analysis of the *E. coli* binary consortium using our model not only showed how interactions between the two amino acid auxotrophic mutants are controlled by the dynamic shifts in limiting substrates but also enabled quantifying previously uncharacterizable complex aspects of microbial interactions, such as asymmetry in interactions. Our approach can be extended to other ecological systems to model their environment-dependent interspecies interactions from growth kinetics.

**IMPORTANCE** Modeling environment-controlled interspecies interactions through separate identification of positive and negative influences of microbes in mixed relationships is a new capability that can significantly improve our ability to understand, predict, and engineer the complex dynamics of microbial communities. Moreover, the prediction of microbial interactions as a function of environmental variables can serve as valuable benchmark data to validate modeling and network inference tools in microbial ecology, the development of which has often been impeded due to the lack of ground truth information on interactions. While demonstrated against microbial data, the theory developed in this work is readily applicable to general community ecology to predict interactions among macroorganisms, such as plants and animals, as well as microorganisms.

**KEYWORDS** microbial communities, competition, cooperation, context dependence, kinetic models, Lotka-Volterra models

M icrobial communities play pivotal roles in maintaining human and animal health, plant productivity, and ecosystem services (1–4). Increasing efforts are being dedicated toward maximizing their beneficial roles in natural systems or creating

Address correspondence to Hyun-Seob Song, hsong5@unl.edu, or Dong-Yup Lee, dongyuplee@skku.edu.

Hyun-Seob Song and Na-Rae Lee contributed equally to this article. Author order was determined by their contributions to theoretical development and manuscript writing.

The authors declare no conflict of interest.

See the funding table on p. 16.

new industrial applications (5). However, control and design of microbial community dynamics and function are challenging tasks, primarily due to higher-order or emergent properties that are not observable from individual species in isolation but arise through nonlinear interspecies interactions (6, 7). Therefore, rational design of microbial communities or consortia requires a fundamental knowledge of microbial interactions as a mechanistic linkage between the environment and the community compositions and function, necessitating the employment of predictive mathematical models as indispensable tools (8–14).

The development of accurate models of microbial communities that are commonly subject to environmental variations is truly complicated by the following intrinsic ecological aspects. First, microorganisms in a community build dynamic interactions that cannot effectively be represented by a rigid network with a fixed structure (15, 16). Rather, microbial communities keep reorganizing interaction networks in response to biotic and/or abiotic perturbations or through adaptation to long-lasting environmental changes. Second, microorganisms often build mixed relationships by exerting both promotive and inhibitive impacts on the growth of their partners/neighbors (17, 18). Individual identification of these simultaneously acting positive and negative interactions is critical because community dynamics is mainly driven by the balances between all counteracting impacts among member species (19). The lack of capability to account for these key properties of microbial interactions limits our ability to predict and engineer microbial community dynamics and functions.

Despite rapid progress in microbiome science, we still do not know how to identify environment-controlled dynamic variation in interspecies interactions addressed above. Three major branches of microbial interaction modeling include (20, 21) (i) network inference, (ii) metabolic network modeling, and (iii) kinetic modeling. Network inference is widely used for modeling microbial interactions to identify interaction networks based on correlative relationships among microbial populations (22–25), parameter identification through regression (26–28), or a prescribed set of rules or hypotheses (21). The resulting networks represent interspecies interactions as single constant metrics, therefore being unable to describe dynamic variations in interactions or identify the balances among counteracting individual impacts in mixed relationships. As an exception, the approach termed minimal interspecies interaction adjustment (15, 16) uniquely enables predicting context-dependent interactions due to the changes in memberships, which, however, has not been extended to address the environmental impacts. In contrast with such data-driven network inference methods, metabolic network and kinetic modeling can account for both positive and negative interactions based on cross-feeding of small molecules (essential for growth) or competition for shared substrates/nutrients among species; in theory, kinetic models can additionally simulate their dynamic variations. While more mechanistic than network inference, these methods cannot quantify the magnitude or even the sign of net interactions.

In this work, we fill these gaps by proposing a novel theoretical framework that enables a quantifiable, mechanistic representation of the dynamic linkage between microbial interactions and the environment. For this purpose, we synergistically integrate two complementary modeling frameworks to overcome their own limitations: a generalized Lotka-Volterra (gLV) model (29) and population growth kinetics. Like other network inference approaches, a typical gLV model with a focus on pairwise interactions is constructed based on an implicit assumption of constant interactions. We relax this assumption by representing interaction coefficients in the gLV model as a function of environmental variables (i.e., concentrations of cross-fed metabolites and shared substrates) described in microbial growth kinetics, which is termed here kinetics-based inference of dynamic variation in microbial interactions (KIDI). The resulting functional representation of interactions by KIDI enables not only quantifying their dynamic variation as environmental conditions change but also individually identifying negative and positive influences among species in mixed relationships. The effectiveness of KIDI was demonstrated through a coordinated design of experiments using a binary

consortium composed of tyrosine and tryptophan auxotrophic mutants of *Escherichia coli* (30) so that they both compete and/or cooperate depending on environmental conditions.

## RESULTS

### Formulation of a conceptual model for understanding environment-dependent interactions

For illustration of the concept of KIDI, we consider a hypothetical consortium composed of two members, where species 1 ($X_1$) and species 2 ($X_2$) cooperate by cross-feeding $S_1^+$ and $S_2^+$ each other but compete for the shared metabolite $S^-$ (the center circle in Fig. 1). Growth kinetics for the $i^{\text{th}}$ species ($X_i$) (which requires two substrates $S_i^+$ and $S^-$ for growth) can be represented, e.g., using a double Michaelis-Menten equation as follows:

$$\mu_i = \mu_i^{\max} \frac{S_i^+}{(K_i^+ + S_i^+)} \frac{S^-}{(K_i^- + S^-)}, \quad i = 1, 2 \tag{1}$$

where $\mu_i$ (1/h) is the specific growth rate of $X_i$, $\mu_i^{\max}$ is the maximal specific growth rate, $s_i^+$ and $s^-$ (g/L) are the concentrations of $S_i^+$ and $S^-$, and $K_i^+$ and $K_i^-$ (g/L) are half-saturation constants associated with the consumption of $S_i^+$ and $S^-$, respectively. As inferable from growth kinetics in equation 1, the mixed relationship (i.e., competition and cooperation) between $X_1$ and $X_2$ when both substrates are limiting can turn into diverse forms of interactions as environmental conditions change. When $S^-$ is present in excess (therefore, no competition is necessary) but $S_1^+$ and $S_2^+$ are limiting, for example, their relationship is predominantly cooperative (where $\mu_i \approx \frac{\mu_i^{\max} s_i^+}{K_i^+ + s_i^+}$). In the opposite case, if both $S_1^+$ and $S_2^+$ are excessive in the environment (so no partners are needed to acquire them) while $S^-$ is limiting, their relationship is governed by competition (where $\mu_i \approx \frac{\mu_i^{\max} s^-}{K_i^- + s^-}$). Likewise, one can assume many other different scenarios where their relationships turn into competition, cooperation, amensalism, commensalism, and even neutrality, as illustrated in Fig. 1.

### Representation of interaction parameters as a function of environmental variables

To model such environment-dependent microbial relationships, we derived a general form of interaction coefficients as a function of environmental variables by integrating growth kinetics and a gLV model. As described in detail in Materials and Methods, our formula (KIDI) represents interaction coefficients of species in the mixed relationship as a sum of positive and negative parts, i.e.,

$$a_{i,j}(s_i^+, s^-) = a_{i,j}^+(s_i^+, s^-) + a_{i,j}^-(s_i^+, s^-), \quad (i, j) = (1, 2) \text{ or } (2, 1) \tag{2}$$

where $a_{i,j}^+$ and $a_{i,j}^-$ denote the positive and negative influence of $X_j$ on the growth rate of $X_i$, which are defined as follows:

$$a_{i,j}^+(s_i^+, s^-) \equiv \frac{\partial}{\partial s_i^+}[\mu_i(s_i^+, s^-)] \cdot \frac{\partial s_i^+}{\partial x_j} \tag{3}$$

$$a_{i,j}^-(s_i^+, s^-) \equiv \frac{\partial}{\partial s^-}[\mu_i(s_i^+, s^-)] \cdot \frac{\partial s^-}{\partial x_j} \tag{4}$$

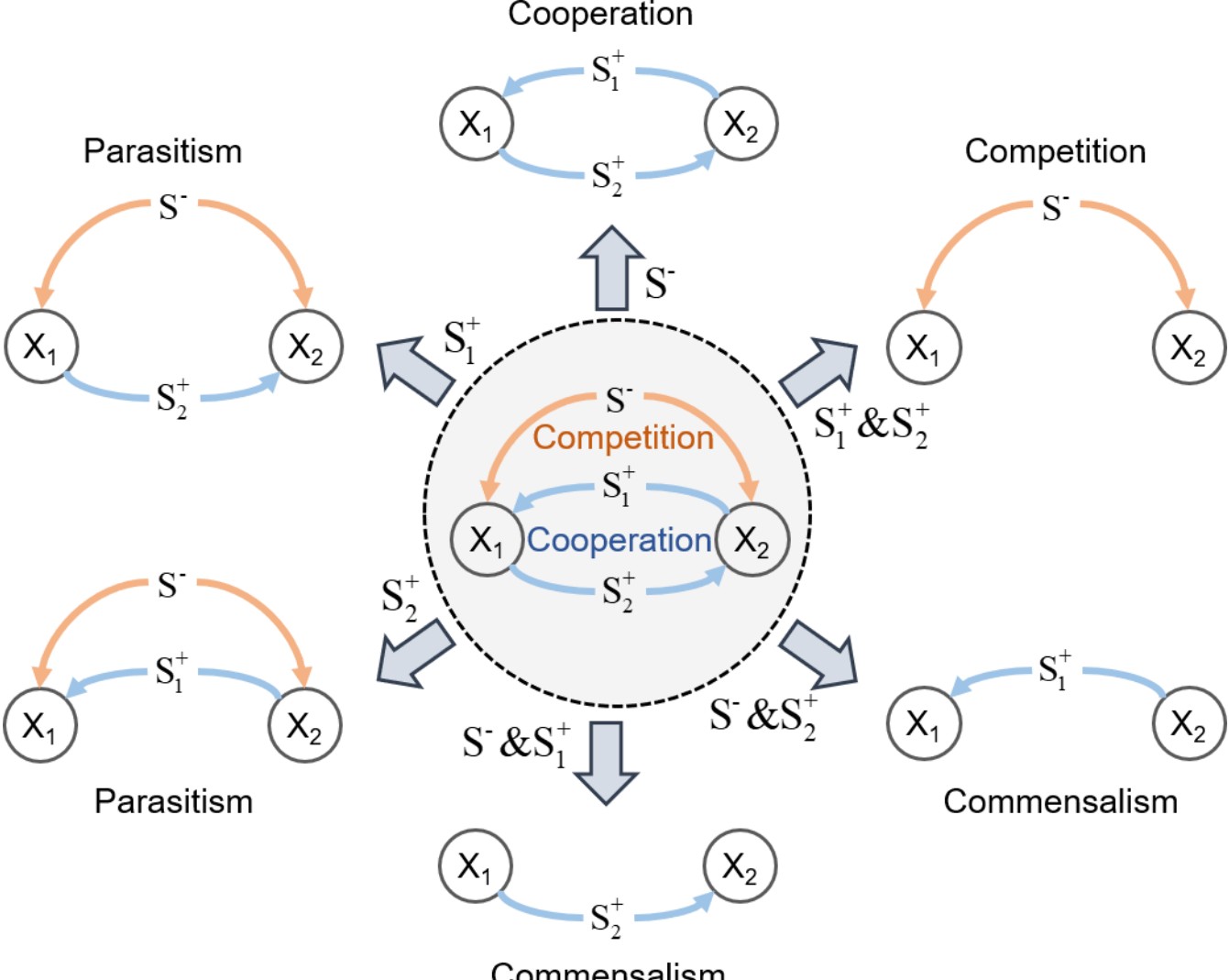

**FIG 1** Conceptual illustration of context-dependent microbial interactions in a binary consortium dictated by the environmental contexts. Two species $X_1$ and $X_2$ compete for the substrate $S^-$ but cooperate by cross-feeding metabolites $S_1^+$ and $S_2^+$ (center panel). This mixed relationship between $X_1$ and $X_2$ diverge into six different types of interactions by excessive addition of specific substrates $S_1^+$, $S_2^+$, and/or $S^-$. Symbols next to the arrows denote the substrate(s) excessively added to the environment.

The positive influence of $X_j$ on the growth rate of $X_i$ (i.e., $a_{i,j}^+$) is represented by the two subsequent terms on the right-hand side of equation 3: (i) the impact of the change in the population size of $X_j$ on the concentration of the cross-fed substrate $S_i^+$ (as denoted by $\partial s_i^+ / \partial x_j$) and (ii) the subsequent impact of the change in $S_i^+$ on the growth rate of the $i^{th}$ species (i.e., $\mu_i$) (as denoted by $\partial [\mu_i(s^-, s_i^+)] / \partial s_i^+$). The negative impact of $X_j$ on the growth rate of $X_i$ ($a_{i,j}^-$) in equation 4 can be interpreted in a similar fashion.

The derivative terms on the right-hand side of equations 3 and 4 are fully identifiable from reaction stoichiometry and kinetics. In the case of using a double Monod kinetics, for example, incorporation of equation 1 into equations 3 and 4 yields $a_{i,j}^+$ and $a_{i,j}^-$ as follows:

$$a_{i,j}^+(s_i^+, s^-) = \left[ \mu_i^{\max} \frac{K_i^+}{\left(K_i^+ + s_i^+\right)^2} \frac{s^-}{\left(K_i^- + s^-\right)} \right] \cdot Y_{S_i^+/X_j} \qquad (5)$$

$$a_{i,j}^-(s_i^+, s^-) = \left[ \mu_i^{\max} \frac{s_i^+}{(K_i^+ + s_i^+)} \frac{K_i^-}{(K_i^- + s^-)^2} \right] \cdot \left( -Y_{S^-/X_j} \right) \quad (6)$$

where $Y_{S_i^+/X_j}$ and $Y_{S^-/X_j}$ denote the stoichiometric relationships between the changes in substrate and biomass concentrations associated with $X_j$, i.e., $Y_{S_i^+/X_j} = \left| \Delta s_i^+ / \Delta x_j \right|$ and $Y_{S^-/X_j} = \left| \Delta s^- / \Delta x_j \right|$ (see Materials and Methods).

To identify net interactions between two species with mixed relationships, we further defined a normalized interaction parameter $\gamma_{i,j}$ as follows:

$$\gamma_{i,j} \equiv \frac{a_{i,j}^+ + a_{i,j}^-}{a_{i,j}^+ - a_{i,j}^-}, \quad (i,j) = (1,2) \text{ or } (2,1) \quad (7)$$

Once $a_{i,j}^+$ and $a_{i,j}^-$ are identified from equations 3 and 4, the parameter $\gamma_{i,j}$ is readily calculable by equation 7. The parameter $\gamma_{i,j}$ ranges from −1 to 1 to represent positive influences of species $j$ on species $i$ when greater than 0 and negative impacts when less than 0, respectively, consequently allowing us to conveniently quantify the relative dominance of inhibition vs promotion in mixed interactions. The parameter $\gamma_{i,j}$ complements $a_{i,j}$, rather than replaces it, in that the magnitude of interactions cannot be determined by $\gamma_{i,j}$, but by the original interaction parameter, $a_{i,j}$. In this regard, $\gamma_{i,j}$ provides an additional complementary explanation of dynamic changes in interspecies interactions. Therefore, all these parameters, including $a_{i,j}$ defined in equations 2 to 4 and $\gamma_{i,j}$, sufficiently characterize the dynamic variation of interactions between $X_1$ and $X_2$ based on co-culture growth data as demonstrated in the following sections.

## Identification of kinetics and stoichiometry via data fit

For experimental demonstration of the mathematical formulation derived in the previous section, we constructed a synthetic consortium composed of two *E. coli* auxotrophic mutants that can cooperatively cross-feed amino acids, while competitively growing on glucose (31). Among 14 amino acid auxotrophic mutants, we chose tryptophan and tyrosine auxotrophic mutants by considering the bioenergetic cost for the synthesis of amino acids based on a previous study in the literature (30). This consortium is considered an ideal, simplest model system for studying environment-dependent dynamic variations in microbial interactions. Due to its exact correspondence to the hypothetical consortium in Fig. 1, we denote two *E. coli* mutant strains ΔtrpC and ΔtyrA by $X_1$ and $X_2$ and glucose, tryptophan, and tyrosine by $S^-$, $S_1^+$, and $S_2^+$, respectively.

Using these two strains, we performed growth experiments under diverse culture conditions: two individual batch experiments using $X_1$ (Fig. 2A) and $X_2$ (Fig. 2B), respectively, and two sets of co-culture experiments (Fig. 2C and D). The top panels in Fig. 2C and D denote co-growth experiments in batch cultures, while the middle and bottom panels denote the same in semi-batch cultures, where we added glucose feedbeads (FBs) at times 7.5 and 10 hours, respectively, to induce dramatic changes in interspecies interactions during co-growth. Other differences in co-culture conditions in Fig. 2C and D include initial concentrations of $S^-$, $S_1^+$, and $S_2^+$, and the number of added FBs (see supplemental material). We measured optical density (OD) at an absorbance of 600 nm as a metric of cell density for the entire culture and determined the relative proportions of the species populations through additional analysis using quantitative PCR (qPCR). By combining these two measurements, we obtained individual ODs for each strain in Fig. 2C. The OD profiles in Fig. 2D denote the combined population change of both strains, i.e., $X_1 + X_2$.

Based on the four data sets in Fig. 2, we constructed a dynamic co-growth model of $X_1$ and $X_2$ to determine associated kinetics and stoichiometry, key information required for quantifying interspecies interaction parameters ($a_{i,j}$ and $\gamma_{i,j}$) in equations 2 to 7. The

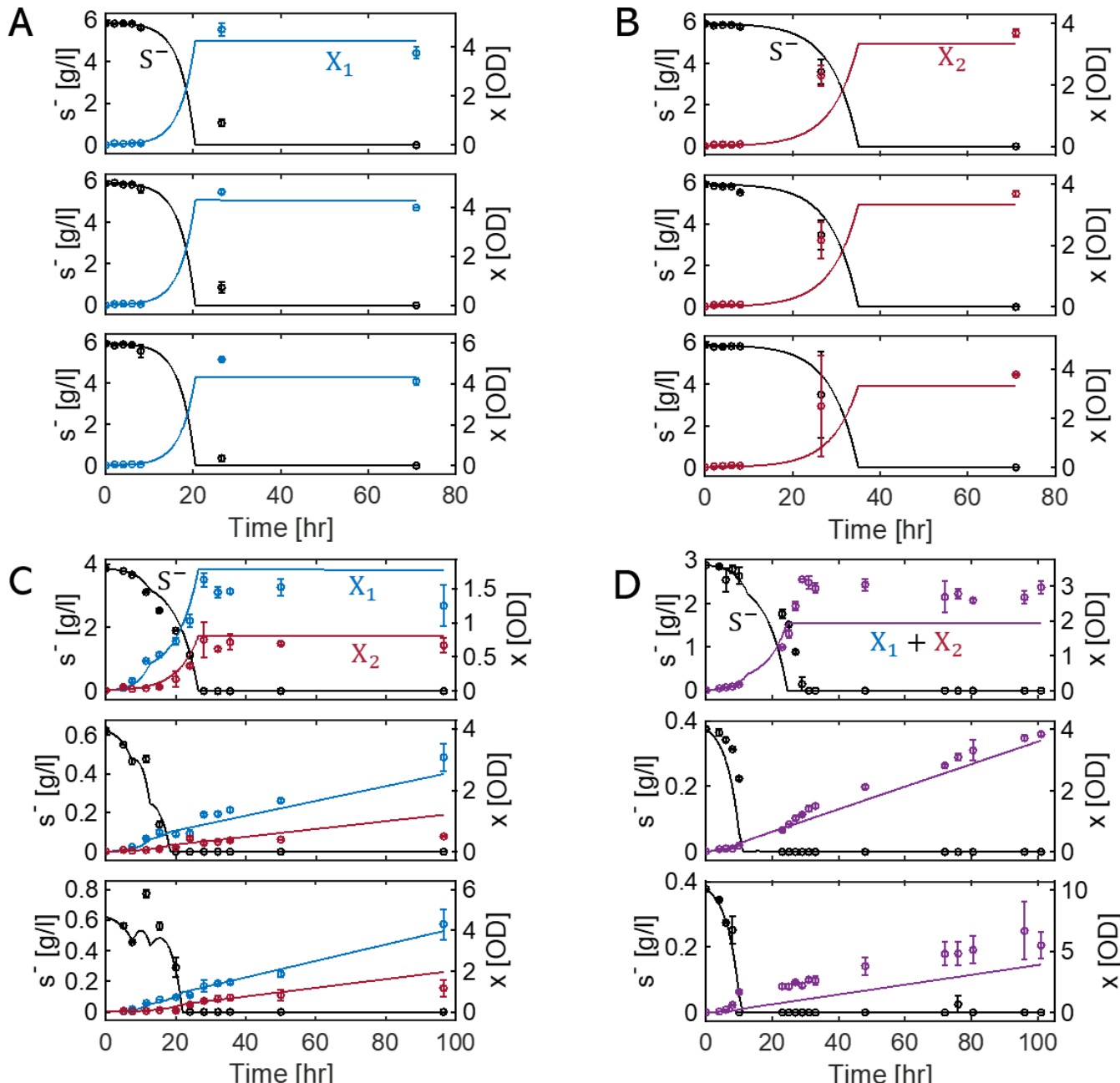

**FIG 2** Experimental data and model simulations for the growth of two *E. coli* mutant strains in axenic and binary culture conditions. (A and B) Cultures of tryptophan auxotrophic and tyrosine auxotrophic *E. coli* mutants ($X_1$ and $X_2$), respectively. (C and D) Co-cultures with two auxotrophs in batch and semi-batch cultures. Detailed culture conditions for the 12 panels are provided in Table S2. Circles and lines denote the experimentally measured values and simulation results, respectively. Black line denotes simulation results for glucose concentration ($S^-$), and the lines in blue, red, and purple are simulated population densities of $X_1$, $X_2$, and $X_1 + X_2$. The vertical error bars represent the standard deviation of measurements across three replicates. Panels A, B, and C show data fitting to determine model parameters, while the results in panel D validate model predictions.

dynamic co-growth model is composed of five mass balance equations for $X_1$, $X_2$, $S^-$, $S_1^+$, and $S_2^+$. We determined stoichiometric and kinetic parameters using three subsets of data in Fig. 2A through C and validated the model against the remaining one (in Fig. 2D) that was not used for model identification. The consistency between simulated and measured data in Fig. 2D, as well as those in Fig. 2A through C, indicates the acceptability of using the identified model parameters in inferring interaction coefficients. The full list

of model equations with parameter values is provided in Table 1 and the culture conditions in Table S1.

While the overall performance of the kinetic model was satisfactory, we found that our experimental setups were not ideal for accurately determining all model parameters. For example, Fig. 2A and B did not effectively show the dependence of the growth of ΔTrp and ΔTry strains on tryptophan and tyrosine because the range of initial concentrations of amino acids (from 10 to 40 mg/L) was too high compared to the half-saturation constants $K_1^+$ and $K_2^+$ (which were determined to be 0.0033 mg/L for tryptophan and 0.00039 mg/L for tyrosine, respectively). This mismatch is partly due to the difficulty in identifying the magnitudes of half-saturation constants in advance before being determined through data fit. We also investigated to what extent model uncertainties could be reduced by expanding the data sets for parameter identification. We found that the inclusion of all experimental data sets from Fig. 2A through D yielded similar parameter values to those listed in Table 1, showing no appreciable changes in both model simulations and KIDI's estimation of interspecies interaction coefficients.

## Variation in microbial interactions driven by the switch in limiting substrates in batch cultures

Based on the stochiometric and kinetic parameters determined through data fit in Table 1, we were able to determine microbial interactions and their variations as a function of environmental conditions using KIDI. We first analyzed various co-culture scenarios in batch reactors (Fig. 3). In Fig. 3A, we considered the growth of $X_1$ and $X_2$ on relatively high and low initial concentrations of $S^-$ (2 g/L) and $S_1^+$ and $S_2^+$ (1 mg/L for both) as a reference condition. In the present setting, the relationship between $X_1$ and $X_2$ is expected to be mostly cooperative (because $S^-$ is excessive in the beginning) and become competitive as the level of $S^-$ decreases. While this overall trend was captured well by our model, the simulation results showed more intricate dynamics than our expectations. As depicted in the top panel of Fig. 3A, the concentrations of $S_1^+$ and $S_2^+$ show a decreasing and increasing trend over time, respectively. This means that $X_2$ does not supply sufficient $S_1^+$ for $X_1$, whereas $X_1$ provides an excess of $S_2^+$ for $X_2$. As a result, the growth pattern of $X_1$ exhibits two distinct exponential phases. Notably, the second phase commences with a slower growth rate when $S_1^+$ becomes depleted in the medium (the second panel from the top). This dynamic accounts for the observed slowdown in the consumption rate of $S^-$ (in the first panel). Overall, these results indicate that the growth of $X_1$ has a greater dependency on $X_2$ than $X_2$ has on $X_1$, particularly when $S_1^+$ becomes depleted. This aspect is correctly captured by the higher values of $\gamma_{1,2}$ than $\gamma_{2,1}$ as shown in the third panel from the top. Actual values of interaction coefficients can be seen from $a_{i,j}$, $a_{i,j}^+$, and $a_{i,j}^-$ (three bottom panels of Fig. 3A), which also showed that $a_{1,2}^+ > a_{2,1}^+$ (and consequently $a_{1,2} > a_{2,1}$) in the second growth phase of $X_2$.

For comparison, we analyzed two additional conditions (i) with a lower initial concentration of $S^-$ (i.e., 0.5 g/L) (Fig. 3B) and (ii) with lower and higher initial concentrations of $S^-$ (0.5 g/L) and $S_1^+$ and $S_2^+$ (i.e., 4 mg/L for both) (Fig. 3C), respectively. Unlike the first case in Fig. 3A, the growth profile of $X_2$ does not show biphasic growth because $S_1^+$ and $S^-$ are depleted almost at the same time. In the case of lowering the initial concentration of $S^-$ (Fig. 3B), KIDI showed that the level of initial competition increases (due to the limited availability of $S^-$) as indicated by relatively lower values of $\gamma_{1,2}$ and $\gamma_{2,1}$ compared to the case of Fig. 3A. Notably, $\gamma_{2,1}$ showed negative value throughout the co-growth (indicating the dominance of negative influence of $X_1$ on the growth of $X_2$). In the case of increasing the initial concentrations of $S_1^+$ and $S_2^+$ in addition to lowering $S^-$ (Fig. 3C), the relationship between the two strains became even more negative (i.e., both

**TABLE 1** Model equations with kinetic parameters and stoichiometric coefficients determined through the model fit to experimental data collected under various limiting conditions[a]

| Equation or parameter |
| --- |

Stoichiometric equation ($R_i$) for the growth of $X_i$

$$R_i : Y_{S^-/X_i}S^- + Y_{S_i^+/X_i}S_i^+ \rightarrow X_i + Y_{S_j^+/X_i}S_j^+, \quad (i,j) = (1,2) \text{ or } (2,1) \tag{T1}$$

Dynamic mass balances

$$\frac{dx_i}{dt} = \mu_i x_i - k_{d,i}x_i, \quad i = 1,2 \tag{T2}$$

$$\frac{ds^-}{dt} = -Y_{S^-/X_1}\mu_1 x_1 - Y_{S^-/X_2}\mu_2 x_2(+q_{S^-}) \tag{T3}$$

$$\frac{ds_i^+}{dt} = -Y_{S_i^+/X_i}\mu_i x_i + Y_{S_i^+/X_j}\mu_j x_j, \quad (i,j) = (1,2) \text{ or } (2,1) \tag{T4}$$

Double Monod kinetics

$$\mu_i = \mu_i^{\max}\frac{s_i^+}{(K_i^+ + s_i^+)}\frac{s^-}{(K_i^- + s^-)}, \quad i = 1,2 \tag{T5}$$

Kinetic parameters and stoichiometric coefficients determined through data fit

| Parameter | Value | Parameter | Value |
| --- | --- | --- | --- |
| $\mu_1^{\max}$ (1/h) | $2.961 \times 10^{-1}$ | $Y_{S^-/X_1}$ (g/OD) | 1.372 |
| $\mu_2^{\max}$ (1/h) | $1.658 \times 10^{-1}$ | $Y_{S^-/X_2}$ (g/OD) | 1.773 |
| $K_1^-$ (g/L) | $3.091 \times 10^{-4}$ | $Y_{S_1^+/X_1}$ (mg/OD) | 1.550 |
| $K_2^-$ (g/L) | $3.923 \times 10^{-4}$ | $Y_{S_1^+/X_2}$ (mg/OD) | 2.961 |
| $K_1^+$ (mg/L) | $3.300 \times 10^{-3}$ | $Y_{S_2^+/X_1}$ (mg/OD) | 1.365 |
| $K_2^+$ (mg/L) | $3.881 \times 10^{-4}$ | $Y_{S_2^+/X_2}$ (mg/OD) | 2.994 |
| $k_{d,1}$ (1/h) | $1.206 \times 10^{-4}$ | $q_{S^-}$ (for three FBs) (g/L/h) | $5.580 \times 10^{-2}$ |
| $k_{d,2}$ (1/h) | $4.024 \times 10^{-7}$ | $q_{S^-}$ (for five FBs) (g/L/h) | $9.300 \times 10^{-2}$ |

[a]$R_i$ is the stoichiometric growth reaction for $X_i$, and $Y_{S^-/X_i}$, $Y_{S_i^+/X_i}$, and $Y_{S_j^+/X_i}$ denote the stoichiometric coefficients for $S^-$, $S_i^+$, and $S_j^+$ associated with the growth of $X_i$. $s^-$, $s_i^+$, and $s_j^+$, respectively, denote concentrations of $S^-$, $S_i^+$, and $S_j^+$, $x_i$ is the population density of $X_i$, $\mu_i$ is the specific growth rate of $X_i$, $k_{d,i}$ is the specific cell death rate of $X_i$, and $q_{S^-}$ is the substrate releasing rate from FBs in glucose-limited semi-batch cultures (i.e., $q_{S^-} = 0$ in a batch mode). $\mu_i^{\max}$ is the maximal growth rate, and $K_i^-$ and $K_i^+$ are half-saturation constants associated with the consumption of $S^-$ and $S_i^+$.

$\gamma_{1,2}$ and $\gamma_{2,1}$ are negative), which was also an expected outcome because metabolic dependence between $X_1$ and $X_2$ will accordingly reduce when they can acquire what they need from the environment, rather than from partners.

In all of these cases, the relations between the two *E. coli* strains were asymmetric, i.e., $\gamma_{1,2} \neq \gamma_{2,1}$, $a_{1,2} \neq a_{2,1}$, $a_{1,2}^- \neq a_{2,1}^-$, and $a_{1,2}^+ \neq a_{2,1}^+$. Asymmetric interactions in terms of $a_{i,j}$'s can also be seen over smaller time windows in Fig. S1. KIDI predicted $a_{1,2} > a_{2,1}$ for the first two cases (Fig. S1A and B) but $a_{1,2} < a_{2,1}$ for the third case (Fig. S1C). In the reference condition where glucose is excessive (so $a_{i,j}^-$'s are relatively negligible), it is mostly due to $a_{1,2}^+ > a_{2,1}^+$ (i.e., $X_1$ has a higher comparative advantage in exchanging amino acids with $X_2$ than the other way around) that leads $\gamma_{1,2} > \gamma_{2,1}$ (as well as $a_{1,2} > a_{2,1}$) (Fig. 3A; Fig. S1A). A similar trend (i.e., $\gamma_{1,2} > \gamma_{2,1}$ and $a_{1,2} > a_{2,1}$) is observed in the second case where all substrates (glucose and amino acids) are limitedly available in the environment and therefore both $a_{i,j}^+$'s and $a_{i,j}^-$'s make comparable contributions to the net interaction coefficients (i.e., $a_{i,j}$'s) (Fig. 3B; Fig. S1B). In contrast with the first two cases, the net interaction coefficients are shown to be $a_{1,2} < a_{2,1}$ for the third case, where the glucose level is low while amino acids are abundant (so $a_{i,j}^+$'s are negligible) because the magnitudes of $a_{i,j}^-$'s are greater than $a_{i,j}^+$'s. Interestingly, KIDI predicted $\gamma_{1,2} > \gamma_{2,1}$ (Fig. 3C) despite $a_{1,2} < a_{2,1}$ (Fig. S1C), which can happen because the implications of $\gamma_{i,j}$ and $a_{i,j}$ are not necessarily identical. The former denotes the relative dominance between

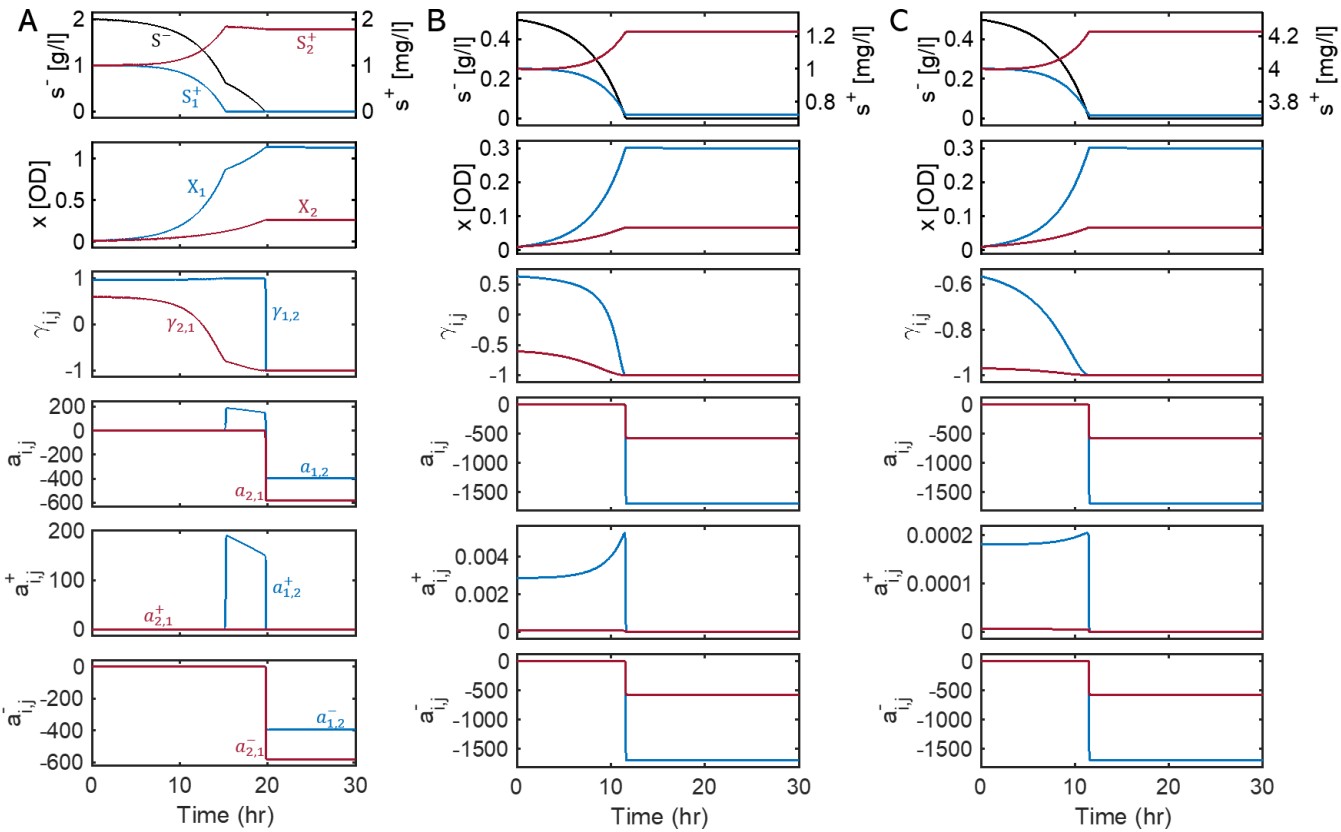

**FIG 3** Inference of dynamic variations of interaction parameters ($\gamma_{i,j}$, $a_{i,j}$, $a_{i,j}^+$, and $a_{i,j}^-$) for the two *E. coli* mutants ($X_1$ and $X_2$) co-growing in three batch cultures. Initial substrate concentrations were (A) 2 g/L of glucose, 1 mg/L of tryptophan, and 1 mg/L of tyrosine; (B) 0.5 g/L of glucose, 1 mg/L tryptophan, and 1 mg/L of tyrosine; and (C) 0.5 g/L of glucose, 4 mg/L of tryptophan, and 4 mg/L of tyrosine. Black line denotes the simulated concentration of glucose ($S^-$); the lines in blue and red indicate the variables and parameters associated with $X_1$ and $X_2$, respectively.

promotion vs inhibition in the relationship of species $i$ with species $j$, while the latter represents the net effect of species $j$ on the growth of species $i$.

## Dynamic response of microbial interactions to environmental perturbations during growth

We extend our analysis to semi-batch cultures that are perturbed by the addition of glucose FBs during growth and therefore are expected to show more dramatic changes in interspecies interactions and community dynamics. In contrast with the batch cultures considered in the previous section, where no further growth is possible after the depletion of the initially added $S^-$, the two strains continue to grow in semi-batch cultures due to slow but continual provision of $S^-$ from the added FBs. Despite a general expectation that the competition level between the two strains will be mitigated at least at the moment of FB addition, it is uncertain (i) to what degree this will occur under different environmental conditions, and (ii) how governing microbial interactions will shift (between competition and cooperation), particularly in a later phase when the growth of the two strains is be limited by both $S^-$ and $S_i^+$. To answer these questions, we applied KIDI to the following three cases. For simplicity, we set the initial conditions to be the same as before.

First, we considered the initial concentrations of 2 g/L for $S^-$ and 1 mg/L for $S_1^+$ and $S_2^+$ and added three FBs of $S^-$ at around 7.5 hours (Fig. 4A). Due to the relatively high concentration of $S^-$, the impact of adding three FBs of $S^-$ on interactions was minimal.

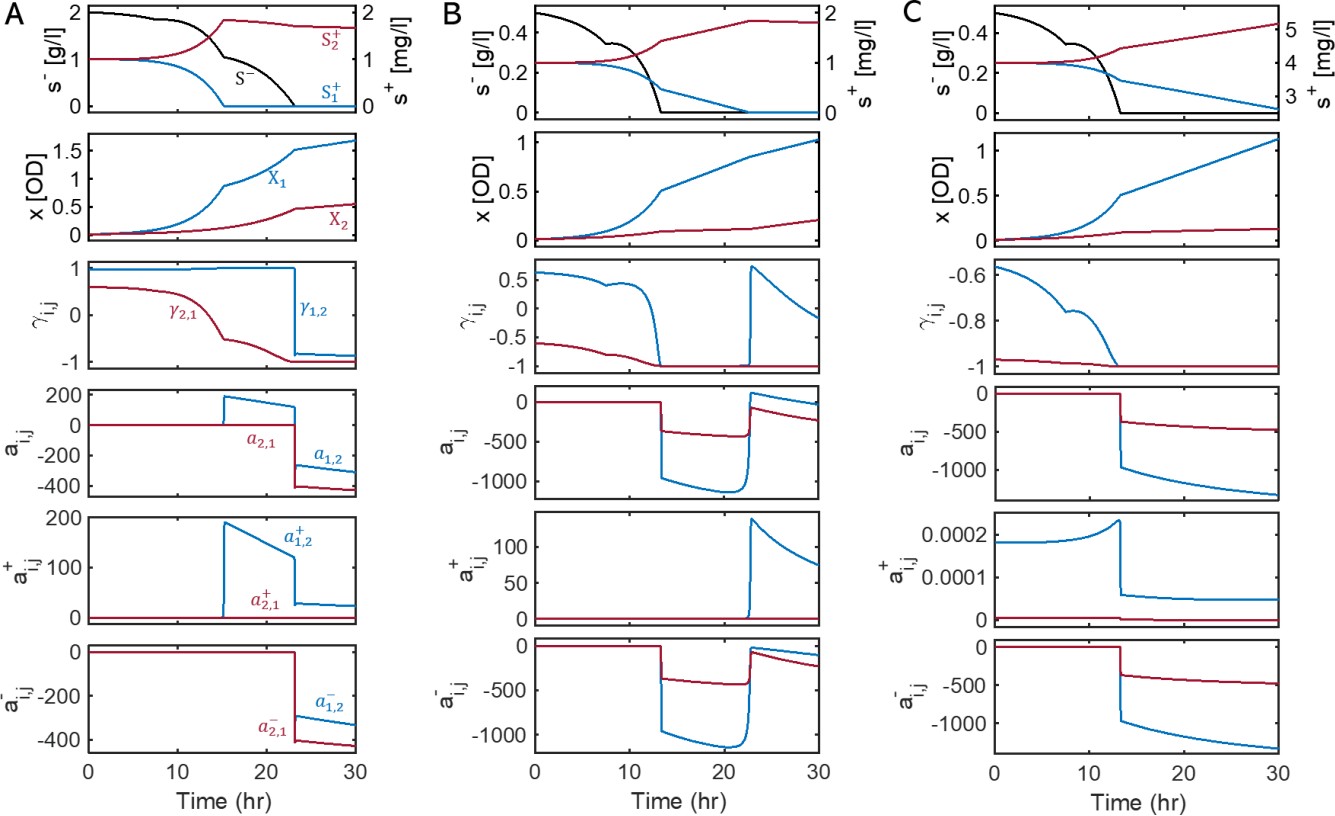

**FIG 4** Inference of dynamic variations of interaction parameters ($\gamma_{i,j}$, $a_{i,j}$, $a_{i,j}^+$, and $a_{i,j}^-$) for the two *E. coli* mutants ($X_1$ and $X_2$) co-growing in three semi-batch cultures with three glucose FBs added at 7.5 hours. Initial substrate concentrations were (A) 2 g/L of glucose, 1 mg/L of tryptophan, and 1 mg/L of tyrosine; (B) 0.5 g/L of glucose, 1 mg/L tryptophan, and 1 mg/L of tyrosine; and (C) 0.5 g/L of glucose, 4 mg/L of tryptophan, and 4 mg/L of tyrosine. Black line denotes the simulated concentration of glucose ($S^-$); the lines in blue and red indicate the variables and parameters associated with $X_1$ and $X_2$, respectively.

The profiles of interaction parameters (i.e., $\gamma_{i,j}$, $a_{i,j}$, $a_{i,j}^+$, and $a_{i,j}^-$ in the four bottom panels in Fig. 3A) as well as the growth curves of $X_1$ and $X_2$ showed no qualitative differences from the batch case (Fig. 3A), while the concentration profile of $S^-$ showed an appreciable increase at the time of addition of three FBs (the top panel in Fig. 3A).

By contrast, when the initial concentration of $S^-$ was low (i.e., 0.5 g/L) (Fig. 4B), KIDI identified the greater impact of adding FBs on both glucose concentration and microbial interactions, as indicated by sudden increases in $S^-$, $\gamma_{1,2}$, and $\gamma_{2,1}$. Interestingly, the value of $\gamma_{1,2}$ shifted to negative (from positive) when the medium was depleted of $S^-$ but reverted to positive upon the depletion of $S_1^+$. The latter suggests a substantial rise in the dependence of $X_1$ on $X_2$ in the absence of $S_1^+$ from the medium. A similar pattern was also noted for $a_{1,2}$, $a_{1,2}^+$, and $a_{1,2}^-$. Additionally increasing the initial concentrations of $S_1^+$ and $S_2^+$ (as shown in Fig. 4C), thereby intensifying the level of competition, resulted in overall patterns similar to the previous case. However, both $\gamma_{1,2}$ and $\gamma_{2,1}$ consistently showed negative values, attributed to the heightened competition. Unlike the previous scenario, there was no increase in interaction parameters, as $S_1^+$ remained available in the medium throughout the time window up to 30 hours. Asymmetry in interaction parameters over a shorter time frame is observable in the detailed views provided in Fig. S2.

The simulations presented in this section demonstrate that the interactions between the two strains are highly nonlinear, influenced by the availability of $S^-$, $S_1^+$, and $S_2^+$ in

the environment, as well as the growth characteristics of $X_1$ and $X_2$. As a general trend, interspecies interactions were dominated by competition when $S^-$ levels were low but shifted toward cooperation in the presence of additional limitations of $S_1^+$ (potentially $S_2^+$ as well). Similar patterns were observed in other scenarios involving perturbations, where both the number of added FBs and the timing of their addition varied, as shown in Fig. S3.

## DISCUSSION

In this study, we proposed a novel computational method (KIDI) that enables quantitatively identifying environment-dependent interspecies interactions in microbial communities. By integrating growth kinetics into a gLV model, we derived an analytical form of interaction coefficients as a function of environmental variables (i.e., concentrations of chemical substrates that affect interactions), the results of which were subsequently validated through a coordinated design of co-culture experiments.

Our theoretical development significantly extends the current scope of microbial ecological modeling by completely relaxing the typical assumption of constant interactions among species. The gLV model, for example, has been widely used as a basic ecological modeling template for the simulation of population dynamics and inference of interspecies interactions in microbial communities (26–28). Due to the constant interaction assumption, however, the application of the gLV model is often confined to a narrow range of conditions where interspecies interactions are expected to remain largely constant. KIDI addresses this limitation by representing interaction coefficients as an explicit function of limiting substrates. As an exception, a previous study by Momeni et al. (32) showed that pairwise interaction (i.e., gLV) models are derivable from mechanistic (i.e., kinetic) models through empirical manipulation of equations, which is, however, limited to special forms of kinetics and therefore cannot be generalizable (32). By contrast, our chain rule-based formulation allows us to handle any complex forms of kinetic equations with no such constraints. Consequently, KIDI enables the incorporation of any form of kinetic equations as demonstrated using a double Michaelis-Menten kinetics as a demonstration example.

Dynamic variations in microbial interactions inferred by KIDI were experimentally validated using a synthetic binary consortium of two metabolically engineered auxotrophic *E. coli* mutants that cross-feed amino acids they cannot synthesize (i.e., tryptophan and tyrosine). A coordinated design of experiments provided multiple sets of data required for determining kinetic and stoichiometric parameters in the mechanistic model along with substrate concentrations, which are key inputs for quantifying environment-dependent interactions. Despite diverse culture conditions including axenic and binary growth in batch and semi-batch reactors, our model with a single set of parameters showed a satisfactory fit to the three training data sets and provided consistency with the validation data set set aside in advance. Such a fair goodness of fit indicates the acceptability of model parameters and, therefore, the subsequent inference of microbial interactions.

Our kinetic model also shows consistency with the analysis of the energetic cost of synthesizing amino acids in the literature. Mee et al. (30) estimated the energetic cost for the synthesis of 14 individual amino acids based on the amounts of extracellularly supplemented amino acids and the observed growth yield of *E. coli* auxotrophic mutants. From the linear relationships between these two variables, they calculated the supplemented amounts of amino acids per cell, which were $1.5 \times 10^7$ and $3.7 \times 10^7$ for the tryptophan and tyrosine auxotrophic *E. coli* mutants, respectively. These two quantities correspond to the stoichiometric coefficients $Y_{S_i^+/X_i}(i = 1,2)$ in our kinetic model, which were determined to be 1.550 ($= Y_{S_1^+/X_1}$) and 2.994 ($= Y_{S_2^+/X_2}$) (mg/OD) through data fit (Table 1). As the direct one-to-one matching between them might not be feasible, e.g., due to different units of biomass [i.e., cell number in Mee et al. (30) vs OD in this work], we compared the ratios, which showed consistency between the two studies, i.e.,

$\frac{1.5 \times 10^7}{3.7 \times 10^7} \approx 0.41$ vs $\frac{1.550}{2.994} \approx 0.52$. Both results imply that compared to tyrosine, the synthesis of tryptophan is more costly. In support of this, Mee et al. (30) estimated that the biosynthetic cost for tryptophan is about 43% higher than that for tyrosine.

We highlight that inferring environment-dependent interactions and their dynamic variations is a critical capability uniquely associated with KIDI. Even in a simple binary consortium considered in this work, KIDI provides new insights into interspecies interactions such as asymmetry between the two amino acid auxotrophs, which might not be obtainable otherwise. In perturbed growth experiments with glucose FBs (as in Fig. 4B), for example, KIDI identified that (i) $a_{1,2}^+ §amp;gt; a_{2,1}^+$ while the shared substrate (glucose) is abundant, implying that the tryptophan auxotroph ($X_1$) does not support the growth of the tyrosine auxotroph ($X_2$) as much as $X_2$ does for $X_1$; (ii) $a_{1,2}^- §amp;lt; a_{2,1}^-$ after the completion of initially added glucose until tryptophan is depleted, implying that less favorable supporters during cooperation become worse enemies when the relationship turned into a competition.

While KIDI determines pairwise interaction terms following the gLV framework, it is also capable of accounting for the influences of additional species, provided these impacts are reflected in the growth kinetics. For instance, in the case study of this article, species 1 and 2 exhibit a complex relationship, competing for glucose uptake while cooperating for amino acid exchange. However, if a third species is introduced, which synthesizes and contributes amino acids to the environment more rapidly than the existing members, the dynamic between species 1 and 2 shifts. Their reliance on each other for amino acids diminishes, transforming their mixed relationship into pure competition due to the influence of the third species.

The chain rule formulation in KIDI successfully estimates interactions from given kinetics, a capability that remains effective across microbial communities of varying complexities. The primary challenge, however, is in identifying growth kinetics. This issue is especially pronounced in complex microbial communities where prior knowledge of interspecies interactions is lacking. Considering these limitations, we showcased KIDI's effectiveness using a binary consortium, which simplified the experimental data collection needed for parameter determination in the mechanistic model. The study of such model microbial consortia, extracted from natural communities, has been instrumental in enhancing our understanding of complex ecological systems (33, 34).

For KIDI to be effectively applied even to simple consortia, comprehensive measurements of all chemical and biological species involved in interspecies interactions are still essential, as precise parameter identification is otherwise challenging. While absent in our study, integrating complete temporal amino acid profiles would improve the accuracy of parameter identification. Typically, metabolite levels exchanged between species, such as amino acids in our case, are low and often fall below detection limits. Additional analysis of axenic culture data would help address this issue.

Despite several challenges mentioned above, it is important to note that these stem from the difficulties in building kinetic models, rather than being a limitation of KIDI itself. KIDI's primary function is to deduce the temporal variations in interaction coefficients based on environmental variables. Its unique ability to handle context-dependent interactions opens up various applications. For instance, KIDI can serve as a probing tool to investigate how assumed growth kinetics and environment-mediated mechanisms lead to specific interactions and their temporal evolution. This aspect is crucial for understanding the link between the growth mechanisms of particular species and their interactions. Moreover, KIDI can greatly aid in advancing network inference techniques. The development of new algorithms for predicting microbial interactions is often hindered by a lack of benchmark data, a gap that KIDI can help fill.

KIDI's utility goes beyond microbial ecology, encompassing a wide range of community ecology fields. This versatility comes from the fact that context dependency is not solely a microorganism trait but is also common in macroorganisms like plants and animals. For example, KIDI is applicable to classic ecological models, such

as MacArthur's consumer-resource model (35). In this model, MacArthur quantifies the impact of consumer $j$ on consumer $i$ ($a_{i,j}$) based on resource population densities and associated parameters, with an underlying assumption that resource populations change more rapidly than consumer populations. KIDI is adept at deducing the competitive relationships among consumers using the chain rule, as detailed in equation (4) (see supplemental material). This adaptability of the KIDI framework enables its extensive use in analyzing context-dependent interactions within and across different biological kingdoms in a variety of ecological systems.

## MATERIALS AND METHODS

### Mathematical definition of interaction coefficients

The dynamic change in population $i$ in a community can be formulated in a general form as follows:

$$\frac{1}{x_i}\frac{dx_i}{dt} = f_i(x_1, x_2, \cdots, x_N), \quad i = 1, 2, \cdots, N \tag{8}$$

where $x_i$ is the population density of species $i$, the left-hand side defines the specific growth rate of species $i$, and the function $f_i(x_1, x_2, \cdots, x_N)$ represents a nonlinear dependence of the specific growth rate of species $i$ on population densities of other species.

Using a Taylor expansion, the right-hand side of equation 8 can be represented as a series of polynomial terms, i.e.,

$$f_i = f_{i,0} + \sum_{j=1}^{N}\left(\frac{\partial f_i}{\partial x_j}\right)_0 x_j + H.O.T., \quad i = 1, 2, \cdots, N \tag{9}$$

where the subscript 0 denotes a chosen reference condition, H.O.T. is higher-order terms. Neglecting the H.O.T. in equation 9, a gLV equation describes the specific growth of species $i$ using a linear equation, i.e.,

$$f_i(x_1, x_2, \cdots, x_N) = f_{i,0} + \sum_{j=1}^{N} a_{i,j} x_j, \quad i = 1, 2, \cdots, N \tag{10}$$

where interaction coefficient $a_{i,j}$ denotes the effect of species population $j$ on the specific growth of species $i$. For a binary community, equation 10 reduces to

$$f_i(x_i, x_j) = f_{i,0} + a_{i,i} x_i + a_{i,j} x_j, \quad i = 1, 2 \tag{11}$$

where $f_{i,0}$ is the basal growth rate of species $i$, $a_{i,i}$ is the intra-specific interaction coefficient, and $a_{i,j}$ is the inter-specific interaction coefficient.

From equation 11, the binary interaction coefficients in gLV are defined as follows:

$$a_{i,j} \equiv \frac{\partial f_i(x_i, x_j)}{\partial x_j} \tag{12}$$

The typical formulation assumes that $a_{i,j}$ is constant, which, however, leads the gLV model to fail to capture the delicate dynamics of microbial interactions. Indeed, $a_{i,j}$ is a dynamic parameter [i.e., $a_{i,j}(t)$] that changes its value in varying environmental conditions as shown in the next section.

## Formulation of interaction coefficients as a function of environmental variables

For simplicity, we assume in this section that $f_i$ in the previous section is represented by kinetic growth rate, $\mu_i$, which is formulated as a function of nutrient concentrations in the environment. In the circumstance considered in Fig. 1,

$$\mu_i(t) = \mu_i[s_i^+(t), s^-(t)] \tag{13}$$

where $s_i^+(t)$ is the concentration of the nutrient (such as tryptophan or tyrosine) at time $t$ that species $i$ needs to get either from its partner or the environment, and $s^-(t)$ represents the concentration of the shared nutrient (i.e., glucose) at time $t$ that two species compete for.

Based on the chain rule, we formulate $a_{i,j}(t)$ as a function of nutrient concentrations by plugging equation 13 into equation 12, i.e.,

$$a_{i,j}(t) = \frac{\partial}{\partial x_j}\{\mu_i[s_i^+(t), s^-(t)]\} = \frac{\partial}{\partial s_i^+(t)}\{\mu_i[s_i^+(t), s^-(t)]\}\frac{\partial s_i^+(t)}{\partial x_j(t)} + \frac{\partial}{\partial s^-(t)}\{\mu_i[s_i^+(t), s^-(t)]\}\frac{\partial s^-(t)}{\partial x_j(t)} \tag{14}$$

Note that the two terms on the R.H.S. of equation 14 represent the positive and negative effects of species $j$ on $i$ through environmental variables, i.e., $a_{i,j}^+$ and $a_{i,j}^-$ as defined below

$$a_{i,j}^+(t) \equiv \frac{\partial}{\partial s_i^+(t)}\{\mu_i[s_i^+(t), s^-(t)]\}\frac{\partial s_i^+(t)}{\partial x_j(t)} \tag{15}$$

$$a_{i,j}^-(t) \equiv \frac{\partial}{\partial s^-(t)}\{\mu_i[s_i^+(t), s^-(t)]\}\frac{\partial s^-(t)}{\partial x_j(t)} \tag{16}$$

In a similar fashion, we can formulate intra-specific interaction coefficients as functions of environmental variables, i.e.,

$$a_{i,i}(t) = \frac{\partial}{\partial x_i}\{\mu_i[s_i^+(t), s^-(t)]\} = \frac{\partial}{\partial s_i^+(t)}\{\mu_i[s_i^+(t), s^-(t)]\}\frac{\partial s_i^+(t)}{\partial x_i(t)} + \frac{\partial}{\partial s^-(t)}\{\mu_i[s_i^+(t), s^-(t)]\}\frac{\partial s^-(t)}{\partial x_i(t)} \tag{17}$$

Final forms of $a_{i,j}(t)$ [$a_{i,j}^+(t)$ and $a_{i,j}^-(t)$] and $a_{i,i}(t)$ depend on specific kinetics for $\mu_i[s^-(t), s_i^+(t)]$. While the symbol $(t)$ is dropped for simplicity, all $a_{i,j}$'s in the main text are dynamic interaction coefficients, the values of which are changing in time as formulated in this section.

## Parameter identification

We determined the optimal parameter values listed in Table 1 by minimizing the sum of squared errors between simulation results and experimental data. During the optimization process, we constrained the half-saturation constants for amino acid consumption and production rates to ensure that the models for the *E. coli* mutant strains could not grow in axenic cultures (Fig. S4) but could co-grow in binary cultures without the external provision of amino acids (Fig. 2C and D).

## Microorganisms and culture conditions

Two auxotrophic *Escherichia coli* (*E. coli*) mutant strains, JW2581-1 and JW1254-2 originally derived from the same strain (BW25113), were purchased from *E. coli* Genetic Stock Center at Yale University (http://cgsc2.biology.yale.edu/). As experimentally validated in the literature (36), these mutant strains, JW2581-1 (ΔtyrA) and JW1254-2 (ΔtrpC), are incapable of growing without supplementation of tyrosine and tryptophan,

respectively. Each strain was incubated overnight at 37℃ and 225 rpm in 50 mL of Falcon tube containing 5 mL of Lysogeny broth supplemented with 33 µg/L kanamycin. Culture cells were collected and centrifuged them at 16,000 × $g$ at 4℃ for 1.5 min. The cell pellets were washed with K3 basal medium to remove residual amino acids in the samples. The washed cells were resuspended and transferred to 150 mL flasks carrying 25 mL of K3 defined minimal medium (5) containing glucose and 33 µg/L kanamycin and cultivated at 37℃ and 225 rpm. An initial absorbance at 600 nm ($OD_{600}$) was 0.04 with an equivalent cell ratio. For batch mode, 4.5 g/L glucose was supplied in the culture medium. For the fed-batch mode, 0.5 g/L of an initial glucose concentration was used to shorten the lag phase, and three or five glucose FeedBeads (Kühner, Basel, Switzerland), releasing glucose at a constant rate, were added when $OD_{600}$ reached 0.2. We collected 500 µL of culture medium from each flask and centrifuged them at 16,000 $g$ for 1.5 min. The supernatant and pellets were stored at −20℃ until further analysis.

## Analysis of glucose concentration in the culture medium

The concentration of glucose was analyzed by a high-performance liquid chromatography system (Agilent, Santa Ciara, CA, USA) equipped with a 1260 refractive index detector and an Aminex HPX-87H column (Bio-Rad, Hercules, CA, USA). Five microliters of filtered supernatants was injected. Analytes were separated isocratically using 5 mM sulfuric acid at a flow rate of 0.7 mL/min.

## Analysis of amino acids concentration in the culture medium

The amino acids in 10 µL of filtered supernatants were analyzed using an ultra-performance liquid chromatography (Waters, Milford, MA, USA) coupled with a micrOTOF II mass spectrometry (TOF-MS) system (Bruker, Bremen, Germany). Analytes were measured using a tunable UV detector at 210 and 397 nm. The amino acids were separated by an Agilent Poroshell 120 EC-C18 column at 30℃. The 1% (vol/vol) of formic acid in water (mobile phase A) and 1% (vol/vol) of formic acid in acetonitrile (mobile phase B) were used, respectively. The amino acids' separation was obtained at a flow rate of 0.3 mL/min with a gradient program that allowed 100% of mobile phase A until 2.1 min followed by increasing mobile phase B to 40% for 2 min and then equilibrated at 0% of eluent B in a total analysis time of 6 min. Analysis of the amino acids was performed using electrospray ionization and full-scan TOF-MS spectra (50–650 $m/z$) with 500 V end plate voltage and 4.5 kV capillary voltage. Nebulizer gas and drying gas were supplied in 1.8 bar and 8 mL/min, respectively. The dry temperature was kept at 220℃.

## Quantification of cell ratio in a microbial consortium

qPCR was carried out in a 96-well plate by using a CFX96 Real-Time Detection System (Bio-Rad, Hercules, CA, USA). The pellets were resuspended in ultra-pure water to make consistent concentration ($OD_{600}$ = 0.4) and then, the 200 µL solution was transferred to a 250 µL PCR tube. The solutions were incubated at 98℃ for 10 min for cell disruption using a T100 Thermal Cycler (Bio-Rad). The lysed cells were transferred to 1.5 mL of tubes and centrifuged at 20,000 × $g$ for 2 min. The supernatants were analyzed by qPCR. The qPCR mixture was composed as follows: 3 µL of 10× Xtensa buffer, 0.3 µL of primer mix (50 µM for each), 0.15 µL of i-Taq (i-DNA Biotechnology, Singapore), 3 µL of 25 mM $MgCl_2$, 5 µL of purified cell lysate, and 18.55 µL of ultra-pure water. The thermal cycling was programmed as follows: 95℃ for 1 min and 30 cycles of 95℃ for 20 s, 55℃ for 20 s, and 68℃ for 40 s. The primers for qPCR analysis to quantify the different *E. coli* strains were provided in Table S3. The qPCR analysis was performed in triplicate for each sample.

## ACKNOWLEDGMENTS

This work was supported by the National Research Foundation (NRF) of Korea grant funded from the Ministry of Science and ICT (MSIT) (No. 2020R1A2C2007192) and the

Korea Institute of Planning and Evaluation for Technology in Food, Agriculture, Forestry and Fisheries (iPET) funded by the MAFRA (32136-05-1- HD050) to D.-Y.L. and by the NRF of Korea grant from the MSIT (No. 2021R1F1A1064592) to S.-Y.P. This material is also based on the work supported by the National Science Foundation under Grant No. 2125155 and Nebraska Tobacco Settlement Biomedical Research Enhancement Funds to H.-S.S.

The manuscript has been released as a pre-print at https://doi.org/10.1101/2022.08.27.505268 (Song et al., 2022).

## AUTHOR AFFILIATIONS

[1]Department of Biological Systems Engineering, University of Nebraska-Lincoln, Lincoln, Nebraska, USA

[2]Department of Food Science and Technology, Nebraska Food for Health Center, University of Nebraska-Lincoln, Lincoln, Nebraska, USA

[3]Research Institute for Bioactive-Metabolome Network, Konkuk University, Seoul, South Korea

[4]School of Chemical Engineering, Sungkyunkwan University, Suwon-si, Gyeonggi-do, South Korea

[5]Department of Chemical and Biomolecular Engineering, National University of Singapore, Singapore, Singapore

## AUTHOR ORCIDs

Hyun-Seob Song  http://orcid.org/0000-0002-2154-6358
Na-Rae Lee  http://orcid.org/0000-0002-1985-0692
Aimee K. Kessell  http://orcid.org/0000-0002-1050-5462
Seo-Young Park  http://orcid.org/0000-0001-6140-412X
Dong-Yup Lee  http://orcid.org/0000-0003-0901-708X

## FUNDING

| Funder | Grant(s) | Author(s) |
| --- | --- | --- |
| National Research Foundation of Korea (NRF) | 2020R1A2C2007192 | Seo-Young Park |
| | | Dong-Yup Lee |
| MAFRA \| Korea Institute of Planning and Evaluation for Technology in Food, Agriculture and Forestry (IPET) | 32136-05-1- HD050 | Seo-Young Park |
| | | Dong-Yup Lee |
| Nebraska Tobacco Settlement Biomedical Research Development Fund (NTSBRDF) | | Hyun-Seob Song |
| | | Aimee K. Kessell |
| | | Hugh C. McCullough |
| National Science Foundation (NSF) | 2125155 | Hyun-Seob Song |
| National Research Foundation of Korea (NRF) | 2021R1F1A1064592 | Seo-Young Park |

## AUTHOR CONTRIBUTIONS

Hyun-Seob Song, Conceptualization, Formal analysis, Funding acquisition, Investigation, Methodology, Project administration, Software, Supervision, Visualization, Writing – original draft, Writing – review and editing | Na-Rae Lee, Conceptualization, Data curation, Formal analysis, Investigation, Methodology, Validation, Writing – original draft, Writing – review and editing | Aimee K. Kessell, Formal analysis, Software, Validation, Writing – review and editing | Hugh C. McCullough, Validation, Writing – review and editing | Seo-Young Park, Validation, Writing – review and editing | Kang Zhou, Conceptualization, Data curation, Resources, Writing – review and editing | Dong-Yup Lee,

Conceptualization, Data curation, Funding acquisition, Methodology, Project administration, Resources, Supervision, Validation, Writing – review and editing

## DATA AVAILABILITY

The data and code for this study are available at https://github.com/hyunseobsong/kidi.

## ADDITIONAL FILES

The following material is available online.

### Supplemental Material

**Supplemental material (mSystems0135-23-s0001.pdf).** Supplemental text; Tables S1 and S2; Fig. S1 to S4.

### Open Peer Review

**PEER REVIEW HISTORY (review-history.pdf).** An accounting of the reviewer comments and feedback.

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
