## [Reviewer comments · mSystems]

Kinetics-based Inference of Environment-Dependent Microbial Interactions and Their Dynamic Variation

Hyun-Seob Song, Na-Rae Lee, Aimee Kessell, Hugh McCullough, Seo-Young Park, Kang Zhou, and Dong-Yup Lee

Corresponding Author(s): Hyun-Seob Song, University of Nebraska-Lincoln

Review Timeline:

Submission Date:	December 5, 2023
Editorial Decision:	January 3, 2024
Revision Received:	March 8, 2024
Accepted:	March 28, 2024

Editor: Gail Rosen

Reviewer(s): The reviewers have opted to remain anonymous.

Transaction Report:

DOI: <https://doi.org/10.1128/msystems.01305-23>

Re: mSystems01305-23 (Kinetics-based Inference of Environment-Dependent Microbial Interactions and Their Dynamic Variation)

Dear Prof. Hyun-Seob Song:

The reviewers are generally supportive. I would add a sentence too on why this study is rigorous enough to claim "robust consistency" in the method with simulation and experimental.

Revision Guidelines

Sincerely,
Gail Rosen
Editor
mSystems

Reviewer #1 (Comments for the Author):

In my opinion, the authors have adequately addressed the concerns raised by reviewers.

Reviewer #2 (Comments for the Author):

Upon resubmission, the theoretical framework described in this paper remains a useful way to combine kinetic (e.g. Michaelis-Menten) and population (e.g. gLV) models, and to decompose bulk interaction parameters into "positive" and "negative" influences of one species on another.

However, in my view the experiments presented in the manuscript (Figure 2) are insufficient to "demonstrate" (lines 13, 90) their framework. Currently the only evidence for KIDI's accuracy is contained in Figure 2d, and in my opinion this plot does not show "robust consistency between simulated and measured data" (line 182), especially given that they are testing on datasets that are relatively similar to the datasets that they used for fitting (glucose levels are similar, and tryptophan and tyrosine concentrations are all much greater than their respective fit half-saturation constants). This is my main concern: I recommend that claims of correspondence between experiment and theory be removed, unless the authors provide additional experimental evidence or implement more quantitative data analysis methods (e.g. sensitivity analysis or Bayesian parameter estimation to demonstrate the inferrability of model parameters).

I am currently unable to critically evaluate the validity of the fits due to the lack of information regarding the parameter fitting procedure; the authors should provide details about their parameter fitting procedures.

I expect that many of the parameters in the author's model (Table 1) are "sloppy", since in the authors' reviewer response letter they say that they modified the parameter set by "re-evaluate[ing] all kinetic and stoichiometric parameters" (without providing details), which in some cases changed parameter values by three orders of magnitude from those in their previous submission. Specifically, I am skeptical of the fit of the tryptophan and tyrosine half-saturation constants, given that these fits are lower (of order $1e-3$ mg/L) by several orders of magnitude than their experimental amino acid concentrations (10 mg/L). I am not convinced that parameters for the KIDI model can be accurately inferred based on the data in Figure 2abc, and I would not call the agreement of the simulation and data in Figure 2d "robust" (line 182).

If parameters are unidentifiable or sloppy, I recommend using the experiments (Figure 2) to motivate a ballpark estimate of parameter values (which could certainly be similar to the parameters the authors currently use). The simulations (Figures 3 and 4) provide interesting case studies with which to examine their KIDI framework.

Lastly, I have a few minor comments:

- + There is inconsistency for the units of glucose (g/l vs mg/l), which makes it difficult to interpret the experimental results: either Table S1 or the Figure 2 y-axis label has a typo, and if the units of glucose truly are g/l, then the fit half-saturation coefficients for glucose K_1^- and K_2^- are also suspect.
- + The $k_{d,i}$ term in Eq T2 is never mentioned in the text.
- + What does it mean that qPCR was used to "determine the OD for each strain" (line 174).
- + What do the error bars in Figure 2 show?
- + 70: what is "unique" about this approach?
- + 144: "calculable"
- + 199: "batch models"
- + 201: "i.e." is not needed
- + line 322: "The model" typo

Reviewer #2 (Comments for the Author):

Upon resubmission, the theoretical framework described in this paper remains a useful way to combine kinetic (e.g. Michaelis-Menten) and population (e.g. gLV) models, and to decompose bulk interaction parameters into "positive" and "negative" influences of one species on another.

[Response] We appreciate the thoughtful comments and feedback from the reviewer. Below, we provide our responses to each of the issues raised by the reviewer.

However, in my view the experiments presented in the manuscript (Figure 2) are insufficient to "demonstrate" (lines 13, 90) their framework. Currently the only evidence for KIDI's accuracy is contained in Figure 2d, and in my opinion this plot does not show "robust consistency between simulated and measured data" (line 182), especially given that they are testing on datasets that are relatively similar to the datasets that they used for fitting (glucose levels are similar, and tryptophan and tyrosine concentrations are all much greater than their respective fit half-saturation constants). This is my main concern: I recommend that claims of correspondence between experiment and theory be removed, unless the authors provide additional experimental evidence or implement more quantitative data analysis methods (e.g. sensitivity analysis or Bayesian parameter estimation to demonstrate the inferrability of model parameters).

[Response] This is a valid point, consistent with previous comments from other reviewers. Although we designed experiments to capture context-dependent interactions carefully, the resulting datasets appear not to be ideal for determining and validating model parameters. Following the reviewer's recommendation, we have removed the claim regarding the robust predictions of the kinetic model. This should not dilute the contribution of this work, because the main strength of KIDI lies in its capability to extract context-dependent interactions from a given kinetic model, as highlighted in the Discussion.

I am currently unable to critically evaluate the validity of the fits due to the lack of information regarding the parameter fitting procedure; the authors should provide details about their parameter fitting procedures. I expect that many of the parameters in the author's model (Table 1) are "sloppy", since in the authors' reviewer response letter they say that they modified the parameter set by "re-evaluate[ing] all kinetic and stoichiometric parameters" (without providing details), which in some cases changed parameter values by three orders of magnitude from those in their previous submission. Specifically, I am skeptical of the fit of the tryptophan and tyrosine half-saturation constants, given that these fits are lower (of order $1e-3$ mg/L) by several orders of magnitude than their experimental amino acid concentrations (10 mg/L). I am not convinced that parameters for the KIDI model can be accurately inferred based on the data in Figure 2abc, and I would not call the agreement of the simulation and data in Figure 2d "robust" (line 182).

[Response] Again, we fully understand this concern and appreciate the reviewer's detailed feedback on the issue of parameter identification. As mentioned in our earlier response, we have now removed the claim of robust predictions for the dataset in Figure 2d. In the Discussion section, we acknowledge the challenges in determining kinetic parameters, especially the half-

saturation constants for amino acid consumption and production. We highlight that this limitation is associated with kinetic modeling in general, rather than KIDI specifically. We have included more details on the parameter identification process in the revised Methods and Materials section.

If parameters are unidentifiable or sloppy, I recommend using the experiments (Figure 2) to motivate a ballpark estimate of parameter values (which could certainly be similar to the parameters the authors currently use). The simulations (Figures 3 and 4) provide interesting case studies with which to examine their KIDI framework.

[Response] As per the reviewer's recommendation, we attempted to redetermine the model parameters based on all datasets in Figure 2. As the reviewer expected, the resulting parameters were very close to those currently used in the manuscript, and there were no appreciable changes in kinetic model simulations and KIDI's estimations of interaction coefficients. Therefore, we decided to retain the current parameter values without having to update all figures and tables. We hope this is acceptable to the reviewer.

Lastly, I have a few minor comments:

+ There is inconsistency for the units of glucose (g/l vs mg/l), which makes it difficult to interpret the experimental results: either Table S1 or the Figure 2 y-axis label has a typo, and if the units of glucose truly are g/l, then the fit half-saturation coefficients for glucose K_1^- and K_2^- are also suspect.

[Response] Thanks for catching this. The unit of glucose should be g/l. The typo in Table S1 has now been corrected. The units of K_1^- and K_2^- are correct as is (i.e., g/l).

+ The $k_{d,i}$ term in Eq T2 is never mentioned in the text.

[Response] $k_{d,i}$ is the specific cell death rate of X_i . We checked that this information is in the caption of Table 2.

+ What does it mean that qPCR was used to "determine the OD for each strain" (line 174).

[Response] We employed qPCR to determine the relative proportions of the species populations (X_1 and X_2) in the cultures. By multiplying their ratios (i.e., X_1/X_2) by the measured OD values for the entire culture, we obtained relative ODs for each strain. For improved clarity, we have revised the text accordingly.

+ What do the error bars in Figure 2 show?

[Response] The error bars represent the standard deviation of measurements across three replicates. We have added this clarification in the revised text.

+ 70: what is "unique" about this approach?

[Response] We referred to KIDI's prediction of context-dependent interactions as the 'unique' capability. To eliminate any ambiguity, we have revised the text accordingly.

+ 144: "calculable"

[Response] Corrected.

+ 199: "batch models"

[Response] We were actually referring to batch modes (i.e., operations). For improved clarity, we have replaced the term 'modes' with 'reactors' or 'cultures' throughout the manuscript.

+ 201: "i.e." is not needed

[Response] Removed.

+ line 322: "The model" typo

[Response] Removed.

Re: mSystems01305-23R1 (Kinetics-based Inference of Environment-Dependent Microbial Interactions and Their Dynamic Variation)

Dear Prof. Hyun-Seob Song:

Your manuscript has been accepted, and I am forwarding it to the ASM production staff for publication. Your paper will first be checked to make sure all elements meet the technical requirements. ASM staff will contact you if anything needs to be revised before copyediting and production can begin. Otherwise, you will be notified when your proofs are ready to be viewed.

Cover Image Submissions: If you would like to submit a potential Cover Image, please email a file and a short legend to msystems@asmusa.org. Please note that we can only consider images that (i) the authors created or own and (ii) have not been previously published. By submitting, you agree that the image can be used under the same terms as the published article. Image File requirements: TIF/EPS, 7.5 inches wide by 8.25 inches tall (at least 2,250 pixels wide by 2,475 pixels tall), minimum 300 dpi resolution (600 dpi preferred), RGB, and no figure elements, e.g., arrows or panel labels. The legend should be a short description of the image, 1-2 sentences recommended.

Sincerely,
Gail Rosen

Editor
mSystems

Reviewer #1 (Comments for the Author):

In my opinion, the adjustments made in response to Reviewer #2's critique has improved the transparency of the manuscript. The contributions and limitations are better clarified in this revised version. Still I believe the manuscript has enough contribution to justify its publication and I think it will stimulate future investigations in this field.

Reviewer #2 (Comments for the Author):

The authors have adequately addressed my concerns.